# Assessment of Management Competencies According to Coherence with Managers' Personalities

**Zuzana Birknerová** [1] and **Ivan Uher** [2,*]

1  Faculty of Management and Business, Univerisity of Prešov, 080 01 Prešov, Slovakia;
   zuzana.birknerova@unipo.sk
2  Institute of Physical Education and Sports, Pavol Jozef Šafárik University, 040 01 Kosice, Slovakia
*  Correspondence: ivan.uher@upjs.sk or ivan.uher57@gmail.com; Tel.: +42-1915-316-532

**Abstract:** The objective of our investigation was to verify a questionnaire's suitability in identifying managerial competencies concerning managers' personality characteristics. Methods: For the content validity of the questionnaire assessment of managerial competencies (AMC23), we investigated its coherence with the appraisal of the management style methodology, i.e., managerial grid (MG), with correlation analysis. The existence of statistically significant relationships between the assessment of managerial competencies and managers' personality characteristics was determined using Pearson's correlation coefficient concerning the BIG-5 model. Results: In total, 573 managers participated in this study. Our examination concludes that motivational competencies correlated positively with the compromise and cooperative style; performance competencies with the competitive and cooperative style; and social competencies with the cooperative, adaptive, and compromise style. Not least, target competencies positively correlated with the competitive and cooperative style. Further, neuroticism negatively correlated with social managerial competence, extraversion, and openness to experience, which positively correlated with motivational and target competence. Friendliness was positively associated with social competence, and diligentness positively correlated with motivational, performance, and target competence. Conclusion: We determined significant correlations between managerial competencies (AMC23) and managerial style (MG). Our findings might have implications for further investigation and the development of more comprehensive instruments to assess managerial competencies in connection with managers' personalities. We point out the need for further research to verify, improve, and constitute a model that further elucidates and explains managerial competencies.

**Keywords:** managerial competencies; manager's personality; managerial style; managerial grade; comprehensive instrument

## 1. Introduction

Management competency is the ability to perform a set of functions and achieve a high level of performance. Managerial competencies are skills that contribute to the excel-lent performance of a managerial role. The theoretical and methodological concept of defining managerial competencies is related to the situational understanding of their application. The recent interest of the professional community in identifying managerial competencies in the context of managers' personalities instigated us to undertake this formative research study, as well as the need to fill the gap on different views on managerial competencies, as represented by the classical situational approach based on dynamic competencies. A different approach is the specification of competencies in terms of personality characteristics that need to be further explored to understand better-related integrands that ultimately enhance managerial performance.

Personnel management was described as early as 1954 by Peter Drucker as an activity set of random techniques. He described it more as a way of life that leads to society's well-being and that of the whole economy [1]. He considered the art of management in

conditions of constant change in order to manage innovatively and with the rapid implementation of innovative strategies being one of the decisive keys to success [2]. It is essential to emphasize the human resource (HR) manager's role [3]; Armstrong [4] delineated HR as the bearer and administrator of organizational values. The HR manager's task is to ensure employees' involvement in corporate culture [5] and to meet the requirements of a market-oriented economy [6].

An organization's performance and competitive advantage can enhance human resource practices [7]. The means by which to increase the efficiency of human resources are the competencies that [8] are characterized as knowledge, skills, abilities, or personality characteristics that affect a person's performance. Furthermore, [9] perceives them as the ability to behave in a way that meets work requirements in the organization's environment parameters while bringing the desired results.

Gibson [10] understands the general competencies as a manager's set of responsibilities and authority. The author's list includes managers' characteristics (e.g., diligence, consistency, self-confidence, charisma, purposefulness, creativity, tolerance, and stability). Among the characteristic features, he adds intelligence, initiative, self-confidence, and enthusiasm. Further, Armstrong and Kim Tran [11,12] defined standard features of managerial competencies. They divided them into behavioral / personnel: interpersonal and analytical skills, leadership, and success orientation; working: professional qualifications, expected performance, and goals in the workplace; species: basic and specific; performance: necessary for the performance of a particular profession; and distinctive: distinguishes high-performing managers from less efficient ones.

The competencies of managers [13,14] are understood as the real skills, knowledge, and experience of a manager, which one should use as effectively as possible to achieve goals. They make a real manager a role model. They distinguish between essential competencies and high-performance competencies, which one incorporates: cognitive, power, motivational, and target competencies. Harris [15] points to creative managers who bring success to organizations that can identify talent and invest in the next generation. Leadership is about impact and change [2]. Therefore, organizations should ensure that the conditions are in place to develop leadership potential [14]. The more leadership styles a manager masters, the more effective it is. One can flexibly adapt the leadership style according to the situation. As Goleman [16,17] claims, it is appropriate to build a team whose members have the styles that a manager needs.

Boyatzis [18] followed up on the research of managerial competencies and researched successful managers' qualities. According to his findings, personality characteristics, such as wisdom, dominance, extraversion, and social and emotional intelligence, explain a significant part of the variance in predicting managers' performance competencies. Moreover, Marques et al. [19] identify engaged leadership, social perceptions, and behavioral flexibility concerning effectiveness.

All managerial characteristics must be understood and interpreted in the overall context of the manager's personality. The manager's character is an integrative component that connects all the above attributes into a single unit. At the same time, it must confront them and apply them in a specific management situation [18].

Armstrong [11] lists various categories of abilities, skills, and qualities that a successful manager should have among the personality characteristics. These can be branch into four areas: knowledge in management, interpersonal skills, conceptual skills and experience, and performance assumptions. By combining them, a specific profile of the manager's personality can be established.

Managers still face the challenge of thinking globally and acting locally [19,20]. Within the manager's profile, personal assumptions from the group of interpersonal skills play a significant role since they have the greatest impact on using the potential that lies in human resources. Among the interpersonal skills, those focusing on the manager's interactions with subordinates are highlighted [21,22]. The main benefit of knowing the manager's per-

sonality is his personality and managerial work. The manager can apply a specific approach to work responsibilities or employees depending on personality characteristics [23,24].

## 2. Material and Methods

Our objective aimed at evaluating managers' competencies in the context of a manager's personality. The primary goal was to verify the questionnaire's suitability to identify the assessment attributes of managerial competencies in the context of managers' personality characteristics.

The research group of 573 managers consisted of 310 (54.1%) men and 263 (45.9%) women aged 20 to 57 years (average age 31.62 years, SD 10.09 years). In terms of classification, there were 104 (18.2%) top, 108 (18.8%) middle, 149 (45.0%) first-level managers, and 212 (37.0%) general directors working in the field of trade services, education, and production organizations located in Slovakia.

To conduct the research, a questionnaire method was used. To evaluate our findings, we employed factor analysis, Cronbach's alpha values, and correlation analysis (Pearson's correlation coefficient) in the statistical program SPSS22. The skewness and kurtosis method determined the skewness and sharpness of the data distribution, and Bartlett's test verified the assumption of the equality of variances in all groups.

The EMC23 (Evaluation of Managerial Competencies 23) questionnaire, aimed at identifying the assessment of managerial competencies, was designed and verified by exploratory and confirmatory factor analysis (Birknerová [25]). The questionnaire evaluates managerial competencies and contains 23 self-judged statements assessed on a 6-point Likert scale (0 = definitely no, one = no, 2 = rather no than yes, 3 = rather yes than no, four = yes, 5 = definitely yes). Based on the results of the Kaiser–Meyer–Olkin Measure of Sampling Adequacy 0.812 and Bartlett's Test of Sphericity 1902.640 (significance 0.000), four factors that characterize managerial competencies were extracted: F1—motivational competencies (examples of items: I know how to motivate others to use their potential. I enjoy inspiring others to perform), F2—performance competencies (examples of items: I like others, I help others to fulfill their tasks), F3—social competencies (examples of items: I want to work with other people. I can quickly adapt to new people), F4—target competencies (examples of items: I can increase people's self-confidence to achieve their goals, I focus my efforts on attaining positive results). Confirmatory factor analysis confirmed the four factors. This study provides an overview of the applicability of the EMC23 methodology in management practice.

Data collection was carried out by quota selection based on information about the distribution of certain characters. The criteria for quotas were: management levels (18.2% top managers, 18.8% middle managers, 45.0% first-level managers, and 37.0% general managers), gender (54.1% men and 45.9% women), and age (average age 31.62 years).

## 3. Results

Four factors were evaluated by factor analysis performed using the principal component method with Varimax rotation, which assesses managerial competencies:

F1 Motivational competencies: Managers with a high score in each factor can build a positive climate and understanding, enjoy motivating others to perform and build their confidence, and inspire people to do what they want. They support others to use their potential and strengths to their benefit and know-how to persuade others to stand by their decisions. The Cronbach's alpha for the F1 factor was 0.881.

F2 Performance competencies: Managers who score highly in each factor help others perform tasks, encourage others in decision making and creativity, advise people about new obligations, are eager to build a team, like to explain the details of a challenging task, and improve performance. The Cronbach's alpha for factor F2 was 0.864.

F3 Social competencies: Managers who score highly in each factor like to cooperate with others, effectively resolve conflicts, communicate with new people, and quickly adapt

to new people interested in other people's values. The Cronbach's alpha for factor F3 was 0.825.

F4 Target competencies: Managers with a high score in each factor can increase the self-confidence of others to achieve their goals owing to their support. Others are dedicated to their work, believe in success, focus their efforts on positive results, and make the most effective decisions to carry out a plan. The Cronbach's alpha for factor F4 was 0.787.

The justifiable Cronbach's alpha values confirm the structure of the extracted factors, representing an acceptable level of reliability of the items within the individual facets of the AMC23 questionnaire. The extracted factors explain 51.366% of the variance, which is a permissible percentage. Based on the sharpness and skew testing, the data distribution in the AMC23 questionnaire factors can be considered normative (Table 1).

**Table 1.** Skewness and sharpness of data distribution in AMC23 questionnaire factors.

|  | Motivational Competencies | Performance Competencies | Social Competencies | Target Competencies |
|---|---|---|---|---|
| Skewness | 0.380 | −0.277 | −0.042 | 0.030 |
| Kurtosis | 0.294 | 0.282 | 0.036 | −0.082 |

Source: own calculations.

We determined the content validity of the AMC23 questionnaire to check coherence with MG managerial styles. The MG methodology [24] consists of two dimensions: performance orientation (focused on production and performance) and people orientation (focused on individuals). It contains nine levels, according to which it is possible to describe five managerial styles (Table 2).

**Table 2.** Managerial styles.

| | |
|---|---|
| Evasive style | low focus on performance and relationships |
| Adaptive style | low performance orientation and high relationship orientation |
| Compromise style | medium focus on performance and relationships |
| Competitive style | high performance orientation and low relationship orientation |
| Cooperative style | high focus on performance and relationships |

Source: authors.

The MG methodology, through which we determined managerial styles, contains 18 items rated on a 6-point Likert scale (0 = never, 5 = always).

Based on peaking and skew testing, the data distribution in the MG methodology can be considered normative (Table 3). A comparison of the results obtained by correlation analysis (Pearson's correlation coefficient) between the factors of the AMC23 questionnaire and the MG methodology is shown in Table 4.

**Table 3.** Summary of the skewness and sharpness of the data distribution in the MG method factors.

|  | Evasive Style | Adaptive Style | Compromise Style | Competitive Style | Cooperative Style |
|---|---|---|---|---|---|
| Skewness | 0.321 | 0.319 | 0.117 | 0.259 | −0.202 |
| Kurtosis | 0.164 | 0.296 | −0.135 | 0.340 | 0.018 |

Source: own calculations.

**Table 4.** Summary of the relationships between the factors of the AMC23 questionnaire and the MG methodology.

| AMC23<br>MG | Motivational<br>Competencies | Performance<br>Competencies | Social<br>Competencies | Target<br>Competencies |
|---|---|---|---|---|
| Evasive style | | | | |
| Adaptive style | | | 0.541 **<br>0.000 | |
| Compromise style | 0.322 **<br>0.000 | | 0.287 *<br>0.012 | |
| Competitive style | | 0.592 **<br>0.000 | | 0.231 *<br>0.025 |
| Cooperative style | 0.491 **<br>0.000 | 0.488 **<br>0.000 | 0.790 **<br>0.000 | 0.636 **<br>0.000 |
| People oriented | 0.564 **<br>0.000 | | 0.601 **<br>0.000 | 0.328 **<br>0.000 |
| Performance orientation | 0.327 **<br>0.000 | 0.754 **<br>0.000 | 0.249 *<br>0.023 | 0.387 **<br>0.000 |

** $p < 0.01$, * $p < 0.05$. Source: own calculations.

Within the AMC23 questionnare, the motivational competencies factor describes managers who can motivate others to use their potential and self-confidence, deliver performance, and inspire them to do what they want. This positively correlates with the MG methodology's compromise style, cooperative style, and orientation towards people and performance. The performance competencies factor describes managers who help others perform tasks, advise people about new and challenging tasks, and encourage them to be creative at work, even in decision making. This correlates positively with the MG competitive and cooperative style and performance-oriented methodology.

The social competencies factor describes managers who like to work with others, effectively resolve conflicts, and communicate with new people. They adapt quickly to others, supporting other people's strengths and values. This correlates positively with the MG methodology's cooperative, adaptive, and compromise style, and the orientation towards people and performance. The target competence factor describes managers characterized by a belief in success, able to persuade others to meet goals, be committed to their work, and have the confidence to support themselves. This positively correlates with the MG methodology's competitive and cooperative style and orientation towards people and performance.

In the relationship between the AMC23 questionnaire and the MG methodology, the cooperative style, which represents cooperation as a collaboration to fulfill common goals and objectives, proved very useful for managers. Statistically significant positive correlation coefficients between the MG methodology factors in the validation study of the AMC23 questionnaire have a high correlation value.

Additionally, we wanted to determine whether there are statistically significant relationships between the assessment of managerial competencies and the personality characteristics of managers using the Big Five model [26,27]. Costa and McCrae [28] performed a series of longitudinal and cross-sectional examinations of the five general dimensions of personality using the NEO inventory. They describe five dimensions of personality (Table 5).

Based on the sharpness and skew testing, it is possible to consider the data distribution in the Big Five model dimensions as normative (Table 6). The correlation coefficients between the factors of the AMC23 questionnaire and the measurements of the Big Five model by the Pearson correlation coefficient in the statistical program SPSS22 are depicted in Table 7.

**Table 5.** Five dimensions of personality.

| | |
|---|---|
| Neuroticism | anxiety, embarrassment, impulsivity, vulnerability |
| Extraversion | warmth, sociability, activity, positive emotions |
| Openness | fantasy, survival, activities, values |
| Friendliness | trust, sincerity, altruism, modesty |
| Diligentness | responsibility, purposefulness, discipline, prudence |

Source: Costa (2008) Source from reference 28.

**Table 6.** Skewness and sharpness of the data distribution in the dimensions of the Big Five model.

| | Neuroticism | Extraversion | Openness to Experience | Friendliness | Diligentness |
|---|---|---|---|---|---|
| Skewness | 0.303 | 0.190 | 0.101 | 0.309 | 0.022 |
| Kurtosis | 0.314 | 0.196 | 0.115 | 0.304 | 0.078 |

Source: own calculations.

**Table 7.** Relationships between AMC23 questionnaire factors and Big Five dimensions.

| AMC23 Big Five | Motivational Competencies | Performance Competencies | Social Competencies | Target Competencies |
|---|---|---|---|---|
| Neuroticism | | | −0.434 ** 0.000 | |
| Extraversion | 0.382 ** 0.000 | | | 0.482 ** 0.000 |
| Openness to experience | 0.377 ** 0.000 | | | 0.389 ** 0.000 |
| Friendliness | | | 0.599 ** 0.000 | |
| Scrupulousness | 0.491 ** 0.000 | 0.347 ** 0.000 | | 0.236 ** 0.020 |

** $p < 0.01$. Source: own calculations.

Personality characteristics such as neuroticism negatively correlate with social managerial competence; extraversion positively correlates with motivational and target competence. Openness to experience positively correlates with extraversion with motivational and target competence, and friendliness positively correlates with social competence and diligent competence. Conscientiousness appears to be a suitable personality characteristic for managers in terms of practical managerial competencies. We found a strong link between friendliness and social skills.

The correlations between self-assessment in individual dimensions of the five-factor model of personality and the assessment of managerial competencies express the degree of agreement between the assessed factors. Our investigation is in line with several findings [11,16–18,27], representing the need to evaluate managerial competencies in the context of the manager's personality.

## 4. Discussion and Conclusions

Based on the results of the Kaiser–Meyer–Olkin Measure of Sampling Adequacy 0.812 and Bartlett's Test of Sphericity 1902.640 (significance 0.000), four factors of the AMC23 methodology were extracted (using the principal component method with Varimax rotation), that assess managerial competencies: F1—motivational, F2—performance, F3—social, and F4—target competencies. Satisfactory values of Cronbach's alpha (F1 0.881, F2 0.864, F3 0.825, and F4 0.787) confirm the structure of the extracted factors. The extracted factors explain 51.366% of the variance, which represents an acceptable percentage.

Determining the content validity of the AMC23 questionnaire was executed with MG methodology using correlation analysis. Motivation thus appears to be a crucial competence in terms of good interpersonal relationships and work performance. Performance competence is beneficial for managers mainly in terms of cooperation and understanding and the execution of demanding tasks. Social competencies are necessary for managers, primarily for support, good relations, teamwork, and practical problem solving. Moreover, target competencies are valued in management to fulfill tasks and for performance and good interpersonal relationships. The team members work towards the same or interrelated goals, listen to each other, complement each other, provide help, and accept responsibility for the result.

We confirmed the suitability of extracting and specifying individual factors of the verified AMC23 methodology focused on evaluating managers' managerial competencies. Pearson's correlation coefficient observed a statistically significant relationship between the assessment of managerial competencies (factors of the AMC23 questionnaire) and the personality characteristics of managers (dimensions of the Big Five model). Managers who appear to be mentally unstable feel nervous or anxious, experience more sadness and fear, cannot resolve conflicts effectively or communicate with people within their competencies, and find it challenging to adapt and cooperate with others. On the contrary, managers who become optimistic, self-confident, sociable, and zealous sometimes alleviate problems within competencies. They can motivate others to perform, inspire them to do what they want, achieve goals, create a favorable climate, believe in success, are dedicated to their work, and have the confidence to support themselves. Ultimately, managers with imagination are curious, independent, prefer change and diversity, and are inclined to new experiences within the competencies that can motivate others to perform, inspire them to do what they want, achieve goals, and create a favorable climate. They also believe in success, are dedicated to their work, and have the confidence to support themselves.

Managers who behave in a friendly manner, are generous and empathetic, trust others, cooperate, are kind, understand, and help, within the competencies, like to collaborate with others. They can effectively resolve conflicts, communicate with new people, adapt to others, and support other people's positive points and values. Managers who appear to be responsible, purposeful, reliable, disciplined, ambitious, diligent, systematic, and persistent with a strong will within the competencies can motivate others to use their potential. Hence, they demonstrate self-confidence, performance, inspiration to do what they desire, how to persuade others to support their decision, and build a positive climate and understanding. They also help others perform tasks, advise people on new and challenging tasks, encourage them to be creative at work and in decision making, and build teams. They attempt to solve problems, clarify strategies and solutions, make the most effective decisions, and look for practical steps to take. A belief in success characterizes them; they can persuade others to achieve goals, be committed to their work owing to their support, and have self-confidence.

Wiseman and Rami [28,29] point to professional self-reflection as an integral part of managerial practice. It is one of the tools that enables continuous personal, professional, and moral growth. Every manager should know their thoughts, feelings, and reasons for their actions in specific managerial situations to move forward and optimize their work. It is significant for a self-reflective manager to explore self-reflective methods and resources [30]. A creative tool for evaluating one's behavior can help answer questions one needs to learn and describe and analyze how to communicate, work, and report to one's surroundings.

One of the specific ways to develop managerial competencies is to use various active training methods that consider the manager's personality [31,32]. The main goal of organizational training is, in addition to the benefits of knowledge, mainly a change in leadership style, corresponding to the organization's intentions that can enhance the managerial practices [33]. In training activities, the emphasis involves experience and awareness of the importance of teamwork and collective decisions. In implementing these activities, it is

necessary to provide feedback concerning the assessment of managerial competencies and managers' personality characteristics [34–37].

Our study exhibits some limitations within which our findings need to be interpreted with caution. The chosen statistical method can be the source of a limitation that emerged during our interpretation of the research and can consequently place constraints on the ability to generalize our results. The research was carried out in Slovakia, i.e., one country. The research presented here was limited by the measures used (business environments are composed of numerous, seemingly uncorrelated, factors that can influence managers' outcomes). Moreover, our study did not examine the somatic characteristics that may have influenced our results and conclusions. The results of our investigation may not be entirely generalizable because the sample size was, within reason, restricted. However, we highlight the significant correlations between managerial competencies (AMC23) and the managerial styles (MG). These findings suggest that managers and executive position candidates should undertake a comprehensive personality test that will evaluate their traits and predispositions to enhance their competence and overall success in the workplace. Our (AMC23) questionnaire can only support this ambition. Our examination may initiate a novel platform and contribute to a new comprehensive instrument to assess the interplay between managerial competencies and managers' personalities. We assert that a higher number of managers with suitable traits can enhance work performance and employees' job satisfaction, and, hence, overall success in the workplace, wherein our assessment can contribute to this ambition.

**Author Contributions:** Study concept and design: Z.B., I.U.; methodology: Z.B., I.U.; formal analysis: I.U., Z.B.; resources: I.U., Z.B.; writing—original draft preparation: Z.B., I.U.; writing review and editing: I.U., Z.B. All authors have read and agreed to the published version of the manuscript.

**Funding:** This work was supported by the project 012PU-4/2020 KEGA: Trading Behavior-Creation of the subject and textbook for non-economic study programs.

**Institutional Review Board Statement:** The study was approved by the Institutional Ethics Committee of the University of Prešov, in Prešov, Slovakia (Approval No. 7.5/2020).

**Informed Consent Statement:** Informed consent was obtained from all subjects involved in the study.

**Data Availability Statement:** The data used to support the findings of this study have been included within the article. The authors have full access to all specific material used in this paper and take responsibility for the use and accuracy of the information provided.

**Conflicts of Interest:** The authors declare no conflict of interest.

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
