# Peer review of "Assessment of Management Competencies According to Coherence with Managers’ Personalities"

_sustainability, doi:10.3390/su14010170_

Round 1

Reviewer 1 Report

Reviewed study presents an interesting research for assessment management competences in the context of the manager's personality. The actuality of the subject is indisputable due to the fact that the manager's figure is the leading one in any organization and therefore the sustainable development of the company depends on the its competences. The paper is logical, well designed completed research showing the results of serious empirical research covering 573 managers. Conclusions and findings also are well justified and reveal the potential for further researches on this subject.

Author Response

Thank you  for your positive commented  we appreciate it very much. Your review encourage as to war even harder to our common goal and that is better science for all. 

Reviewer 2 Report

The topic is interesting but considerable work must be conducted before the article be considered for publication. The changes that must be conducted by the authors are:

Introduction. Is lacking. The authors go directly to the literature review without presenting the reason for this study, the research gap, objectives and the methodological approach.

About the literature review: is old-fashioned with most of the references before 2015. It is also insufficient. I deeply recommend including references listed as Q1 in Scopus or WoS. For example, what is the relevance of this source in a journal as Sustainability: Hřebíčková, M; Urbánek, T. NEO Five-Factor Inventory. Translation from: NEO Five-Factor Inventory Costa PTand McCrae 341RR). Praha: Testcentrum; 2001?

I recommend that the authors should revise the theoretical framework using more recent and relevant articles

Method: This is the principal weakness of the article. Please consider more advanced approaches like structural equation modeling. 

Furthermore, the authors reveal that "To conduct research questionnaire method was used.". Where is the questionnaire? Where do the questions come from? How were they validated?

About the respondents: describe the procedures regarding data collection.

Author Response

Dear reviewer,

we responding to your comments below.

  1. Introduction. The authors go directly to the literature review without presenting the reason for this study, the research gap, objectives, and the methodological approach.

It Has been added.

  1. For example, what is the relevance of this source in a journal as Sustainability: Hřebíčková, M; Urbánek, T. NEO Five-Factor Inventory. Translation from: NEO Five-Factor Inventory Costa PTand McCrae 341RR). Praha: Testcentrum; 2001?

The questionnaire, Hřebíčková, M; Urbánek 2001. Represents a recognized psychological questionnaire in our background (high validity and reliability) focused on assessing personality characteristics. Therefore, we choose to work with it.

  1. I recommend that the authors revise the theoretical framework using more recent and relevant articles.

It has been revised and adding five more relevant articles.

  1. Method: This is the principal weakness of the article. Please consider more advanced approaches like structural equation modeling.

Even though SEM is more powerful than regression analysis to fulfill our research goal, we have used: Cronbach's alpha, correlation analysis (Pearson's correlation coefficient), Skewness and Kurtosis method, Kaiser-Meyer-Olkin Sampling Adequacy test, Bartlett's Test of Sphericity in the SPSS22 statistical program.

  1. Furthermore, the authors reveal that "To conduct research questionnaire method was used.". Where is the questionnaire? Where do the questions come from? How were they validated? About the respondents: describe the procedures regarding data collection.

It has been added.

Thank you once more for your time and effort to a common goal.

With respect

Ivan Uher

Round 2

Reviewer 2 Report

Dear authors

I'm not satisfied at all with the revision. The answers you gave are not worthy of a good quality journal as Sustainability. I'm returning the revision back. Please inform yourselves how to conduct a revision report.  "It Has been added." is no response for a reviewer. What specific changes did you make it?

This is no response to a reviewer: "Even though SEM is more powerful than regression analysis to fulfill our research goal, we have used: Cronbach's alpha, correlation analysis (Pearson's correlation coefficient), Skewness and Kurtosis method, Kaiser-Meyer-Olkin Sampling Adequacy test, Bartlett's Test of Sphericity in the SPSS22 statistical program.". The minimum acceptable was to recognize the weak approach in the limitations. I was expecting that you consult previous research (published in Scopus Q1 journals) using the same methodological approach to defend your position.

The same goes for this answer "The questionnaire, Hřebíčková, M; Urbánek 2001. Represents a recognized psychological questionnaire in our background (high validity and reliability) focused on assessing personality characteristics. Therefore, we choose to work with it." Please provide better arguments or conduct additional validity for a scale that was not published in a Scopus Q1 as Sustainability. 

Author Response

Dear reviewer,

Below sending comments to your review.

1.

"The questionnaire, Hřebíčková, M; Urbánek 2001. Represents a recognized psychological questionnaire in our background (high validity and reliability) focused on assessing personality characteristics. Therefore, we choose to work with it." Please provide better arguments or conduct additional validity for a scale that was not published in a Scopus Q1 as Sustainability. 

  1. The personality concept (BIG-5) has a practical application in working psychology and management. We attempt to employ (BIG-5) concept in the context of evaluation of managerial competencies. We aimed to determine whether there is a statistically significant correlation between the BIG-5 questionnaire and our EMC23 questionnaire., BIG-5 questionnaire focuses on determining personality characteristics. Our questionnaire focused on managerial competencies.

Below is a presented summary of the research paper “Lexical and dispositional approach to the five-factor model” by prof. Martina Hřebíčková DrSc. (Doctor of Science) presented at the Academy of Science of Czech Republic in Prague 2008 that analyze BIG-5 model concerning its substantiation. 

Summary:

One of the most dynamic areas of personality research during the past two decades has been that of personality structure. The structure of personality characteristics has been examined using the lexical strategy for finding the major personality dimensions. The rationale for lexical studies rests on the assumption that the most meaningful personality attributes are encoded in language as single-word descriptors. Based on this rationale, a number of studies have been conducted examining the factor structure mainly of adjectival descriptors extracted from dictionaries. The results of many studies in the field have supported the validity of

the „Five-Factor Model“ with factors identified as (1) SURGENCY or EXTRAVERSION (talkative, assertive, energetic), (2) AGREEABLENESS (good-natured, co-operative, trustful), (3) CONSCIENTIOUSNESS (conscientious, responsible, orderly), (4) EMOTIONAL STABILITY or its opposite NEUROTICISM (calm, neurotic, not easily upset), (5) CULTURE, INTELLECT or in one inventory representation OPENNESS TO EXPERIENCE (cultured, intellectual, unconventional). As the Five-Factor model has been shown to be robust across a diversity of studies, the five factors have also been

called the Big Five.

In the framework of the Big Five Model, two approaches are usually differentiated: lexical (taxonomic) and dispositional (questionnaire). The aim of the lexical research of the five-factor model of personality is to organize personalityrelevant words and to find the most important of them. Therefore it is a model of personality attributes that is descriptive rather than explanatory. In contrast, the Five-Factor Model in a dispositional approach is based on factor analyses of questionnaire scales. It is assumed that the five factors correspond to biological traits, that can explain behavior. Both these approaches have been implemented in

research on the Five-Factor Model in Czech.

A set of 26 published scientific publications is a basis of my dissertation. All articles and chapters published between 1993 – 2007 relate to five-factor model of personality. 11 studies concern the lexical approach in the Big Five and 15 studies represent the dispositional approach to the Five-Factor model. In the first part of the Thesis, I present the overview of the lexical studies in the Czech context. First, the lexical analysis of the Czech personality-relevant adjectives is being described. The studies confirmed the generalizability of the Five- Factor model in the Czech language as well as the robustness of the Big-Five across different samples of variables, rating inventories, and groups of raters. Thus, the major factors of the Czech personality language could be interpreted as the Big Five (Hřebčkov., 1997, 1999).

Despite of the fact that results of the most lexical studies have supported the validity of the Big-Five model, the number and content of the factors are still discussed. One possible way how answer the question concerning the content of the five-personality dimension is to compare the five-factor solutions in national representative five-factor structures. The Czech five-factor structure was compared with Dutch, German, Italian, and Polish Five-Factor structures (DeRaad et al, 1998). The results show that Hungarian and Czech are least accommodated in American English trait structure. The first three factors (I-III) of the six national five-facto structures come close to strict similarity. With an exception of German, the American English Factor V cannot be identified in closely similar form in any of the other

languages. The situation for Factor IV is somewhat better then for factor V.

The 5-dimensional simple structure contains traits that loaded only on one factor and secondary loadings are ignored. The Abridged Big Five Dimensional Circumplex (AB5C) taxonomy integrates simple-structure and circumplex models. In AB5C every trait is characterized by its position to two other factors of the five-factor structure, thus containing 10 two-dimensional circumplexes. Hřeb.čkov. and Ostendorf (2005) applied AB5C methodology to data consisting 397 self-ratings on 358 Czech representative personality trait adjectives. From theoretically possible 45 bipolar facets, 26 can be defined on the basis of the Czech data analysis. By integrating the simple-structure and circumplex approaches to trait structure, the AB5C procedure should provide a general framework into which many earlier

conceptions can be fitted and through which the relations among these conceptions

may be clarified.

At the end of the lexical part of the thesis, the taxonomy and structure of the Czech personality-relevant verbs is reported. Until recently, most taxonomies were based on analyses of personality-descriptive adjectives only. Our lexical study was the first attempt to select all personality-relevant verbs from the Czech lexicon (Hřebčkov., et all, 1999). The resulting comprehensive and representative list of Czech personality-relevant verbs was a tool for the examination of the major dimensions of personality description. In certain respects, the structure of personality descriptive verbs resembles the structure of adjectives found in the personality lexicon. The first verb factor refers to characteristics that are summarized by the Big- Five factor II (Agreeableness) in the domain of adjectives. The second verb factor includes characteristics of the Big-Five factors Emotional Instability, Introversion and Extraversion. The fourth verb factor of the four-factor solution seems to be a parallel to Conscientiousness, the well known Big-Five factor III. The most important difference between the structures of the two word classes seems to be that there is no verb factor covering the content of Big-Five factor V, Intellect or Openness to Experience.

The dispositional (questionnaire) approach to the Five-Factor Model has been elaborated primarily by Costa and McCrae. Their Five-Factor Theory of personality and their instrument, the NEO Personality Inventory, were originally developed in the context of longitudinal studies of personality and aging. The Czech version of the instrument (NEO-FFI and NEO-PI-R) were used in several research projects and their results are included in the published articles.

The short version of the NEO-Five-Factor Inventory (NEO-FFI) has been translated into Czech, Polish and Slovak and the psychometric characteristics were examined (Hřeb.čkov. et all, 2002). We assumed that people from countries who speak similar languages and live in similar cultural conditions would not differ in the level of personality traits. Our results showed otherwise. Among Czech, Slovak, and Polish adolescents there are statistically significant differences. Slovak adolescents provide most extreme self-reports – they are more emotionally stable, extraverted, and conscious then Czech and Polish adolescents. Polish adolescents describe themselves as more open in comparison with Czech and Slovak adolescents. In the cross-cultural studies assessing changes in personality traits during life,

the Czech data were used as well. McCrae and coworkers compared personality traits of 5,085 respondents aged from 14 till 80 years from five countries (Germany, United Kingdom, Spain, the Czech Republic, and Turkey) (McCrae, et all, 2000). The results confirm that there is a significant decline in Neuroticism and Extraversion and increase in Conscientiousness in all five samples. A significant increase in Agreeableness and decline in Openness to Experience wasn’t found in all five samples. The developmental trends were confirmed here on the basis of the selfrating using the NEO-FFI. The developmental trends of five general personality traits

were studied not only in the self-reports but also in peer-ratings using the NEO-PI-R in Czech and Russian respondents (McCrae et all, 2004a). The analysis of self-rating from both countries confirms the developmental trends found earlier – Neuroticism, Extraversion, and Openness to Experience decline while Agreeableness and Conscientiousness increase with age. In the Russian sample, a decrease of Extraversion and Openness and an increase in Agreeableness and Conscientiousness with age was confirmed by peer-rating; in Czech subjects, the identical trend was confirmed by peer-rating only for Extraversion and Openness. The findings of the identical developmental trends in respondents coming from countries with different histories, development, religion, culture, and language was very important for the authors of the Five-Factor theory because they confirm their assumptions. It is the intrinsic maturation supposed to be universal for all humans and neither the cultural influence nore the environment that explains the developmental trends recorded.

In a framework of the dispositional approach to the five-factor model, a cross-observer agreement on personality trait ratings has been studied as well. I deal with methodological issues of research in self-other agreement (Hřeb.čkov., 2003). A cross-cultural empirical study was published on this topic by McCrae and colleagues (McCrae et all, 2004b). In the Czech and Russian sample, all five factors showed clear evidence of inter-judge agreement. It appears that facets of Extraversion: Gregariousness, Assertiveness, Activity, and Excitement Seeking are easily rated by observers, presumably because they involve easily observed behaviors. The evidence that spouse-raters are generally superior to other raters (e.g. friends, siblings) in a self-other agreement was replicated only in the Russian data. In the Czech data,

other relatives or friends show higher self-other agreement in comparison with spouse.

Dimensional or variable-centered approaches describe an individual’spersonality by its scores on several personality dimensions. The person-centered research aims at identifying types of individuals that share the same basic personality profile. In various studies, three major personality types could consistently be identified. The most often mentioned personality types are “Resilients” (low scores in Neuroticism and above average scores an all other scales), Overcontrollers (low scores in Extraversion and high scores in Neuroticism) and Undercontrollers (low scores on Agreeablenss and Conscientiousness). Hřebčakov. and Urbanek (2006) deal with the question of whether the three major personality types can be replicated in the Czech sample. Cluster structure in self-report based on the NEO personality

Inventory (NEO-PI-R) was examined in a sample of 2,279 Czech subjects aged from 14 to 83 years. The highest replicability was identified for the three-cluster solutions, but the four-cluster solution was replicated as well. Only Resilient type could be clearly replicated in the three-cluster solution, whereas typical Big Five patterns of the Overcontrolled and Undercontrolled prototypes mixed into two types.

The Five-Factor model of personality-description based on the tradition of lexical studies is still a working hypothesis that can be used in further research of terms relevant for personality structure. However, the Five-Factor model of personality represented by NEO Inventories has been already widely applied in research studies as well as in psychodiagnostics.

  1.  

I'm not satisfied at all with the revision. The answers you gave are not worthy of a good quality journal as Sustainability. I'm returning the revision back. Please inform yourselves how to conduct a revision report.  "It Has been added." is no response for a reviewer. What specific changes did you make it?

Incorporated changes includes: Enhancement of Introduction 35-44., Closer specification of our questionary 134-153., Modification and Enhancement of 18 literary sources 333-397.

3.

This is no response to a reviewer: "Even though SEM is more powerful than regression analysis to fulfill our research goal, we have used: Cronbach's alpha, correlation analysis (Pearson's correlation coefficient), Skewness and Kurtosis method, Kaiser-Meyer-Olkin Sampling Adequacy test, Bartlett's Test of Sphericity in the SPSS22 statistical program.". The minimum acceptable was to recognize the weak approach in the limitations. I was expecting that you consult previous research (published in Scopus Q1 journals) using the same methodological approach to defend your position.

The chosen statistical method can be the source of limitation that emerged during our interpretation of the research and can consequently place constraints on the ability to generalize our results. 311

Thank you for your valuable and stimulating comments on our contribution, which will help us broaden our perspective in the context of the paper and beyond.

We apologize for the extra time imposed on you by our inoperativeness.

Thank you once again.

With respect

Ivan Uher

Round 3

Reviewer 2 Report

.